# Cortical representations of numbers and nonsymbolic quantities expand and segregate in children from 5 to 8 years of age

**Tomoya Nakai** *, **Cléa Girard, Léa Longo, Hanna Chesnokova, Jérôme Prado** *

Lyon Neuroscience Research Center (CRNL), INSERM U1028—CNRS UMR5292, University of Lyon, Bron, France

* nakai.tomoya@neuro.mimoza.jp (TN); jerome.prado@univ-lyon1.fr (JP)

**Data Availability Statement:** The scripts and individual MRI data are available from Zenodo (https://doi.org/10.5281/zenodo.7285545).

## Abstract

Number symbols, such as Arabic numerals, are cultural inventions that have transformed human mathematical skills. Although their acquisition is at the core of early elementary education in children, it remains unknown how the neural representations of numerals emerge during that period. It is also unclear whether these relate to an ontogenetically earlier sense of approximate quantity. Here, we used multivariate fMRI adaptation coupled with within- and between-format machine learning to probe the cortical representations of Arabic numerals and approximate nonsymbolic quantity in 89 children either at the beginning (age 5) or four years into formal education (age 8). Although the cortical representations of both numerals and nonsymbolic quantities expanded from age 5 to age 8, these representations also segregated with learning and development. Specifically, a format-independent neural representation of quantity was found in the right parietal cortex, but only for 5-year-olds. These results are consistent with the so-called symbolic estrangement hypothesis, which argues that the relation between symbolic and nonsymbolic quantity weakens with exposure to formal mathematics in children.

## Introduction

Learning Arabic numerals is a milestone in early elementary education. It is also the first step towards understanding symbolic mathematics, which is fundamental for academic growth in children. Prior studies suggest that brain sensitivity to approximate nonsymbolic quantities precedes the acquisition of number symbols in children [1,2], with early neural processing of approximate nonsymbolic quantities reported around the intraparietal sulcus (IPS) [1–4]. This is consistent with the idea that an early-developing "approximate number system" (ANS) [5] may scaffold the development of symbolic numerical knowledge. However, very little is known about (i) how neural representations of Arabic numerals emerge in the first years of schooling and (ii) how these representations relate to neural representations of approximate nonsymbolic quantities in children.

Overall, neuroimaging studies that investigated separately symbolic and nonsymbolic quantity processing have identified similar networks encompassing the IPS and prefrontal cortex in

**Funding:** This study was funded by the Agence Nationale de la Recherche (ANR-14-CE30–0002 and ANR-17-CE28–0014) to J.P., and by JSPS Overseas Research Fellowship to T.N. The funders had no role in study design, data collection and analysis, decision to publish, or preparation of the manuscript.

**Competing interests:** The authors have declared that no competing interests exist.

**Abbreviations:** AAL, automated anatomical labeling; ANS, approximate number system; BOLD, blood oxygenation level-dependent; FG, fusiform gyrus; fMRI, functional magnetic resonance imaging; IFGtri, triangular part of inferior frontal gyrus; IPL, inferior parietal lobule; IPS, intraparietal sulcus; IQ, intelligence quotient; LOOCV, leave-one-out cross-validation; MFG, middle frontal gyrus; MNI, Montreal Neurological Institute; PreCG, precentral gyrus; SD, standard deviation; SES, socioeconomic status; SPL, superior parietal lobule.

normal adults (for a meta-analysis, see [6]). This is consistent with a few studies that have also reported shared activation or decodability when quantities are presented to the same individuals in different formats [7–9]. Such shared parietal activity has also been observed in a small group of 6- and 7-year-old children [10]. Together, these studies support the idea that there is an abstract (i.e., format-independent) representation of numerical quantity in the brain, with similar neuronal populations coding for symbolic and nonsymbolic quantity [11].

However, a growing number of studies have also failed to find an overlapping neural activity for symbolic and nonsymbolic quantity processing [12,13]. For example, two recent reports have shown distinct neural representations of number symbols and nonsymbolic quantities across the largest samples of adult participants to date [14,15]. These studies suggest that quantity may be represented in a format-dependent manner in the brain [16]. This is consistent with the idea that the emergence of symbolic number knowledge in humans mainly results from cultural practices that have more to do with mastering the logic of counting than mapping symbols onto a perceptual sense of quantity [16–18].

A source of difficulty in interpreting the results of previous studies is that they largely focus on educated adults. Educated adults have been exposed to symbolic numbers for many years and have thus acquired extensive experience manipulating symbols without referring to the quantity they represent. Such experience might significantly weaken any preexisting relation between number symbols and nonsymbolic quantity [19]. In other words, there might still exist a relation between the neural representations of symbolic and approximate nonsymbolic quantity in young children, as would be expected if the ANS scaffolds symbolic numerical skills. However, this relation might disappear with exposure to formal mathematics through elementary school.

In the present cross-sectional study, we used functional magnetic resonance imaging (fMRI) to investigate the emergence of the neural representations of Arabic numerals from age 5 to age 8, which corresponds to the first four years of formal education in children. We also aimed to assess the relation between the representations of Arabic numerals and approximate nonsymbolic quantity during that period. Specifically, we made four hypotheses regarding the relation between the representations of Arabic numerals and approximate nonsymbolic quantities in 5- and 8-year-olds. These representations could be (1) always similar; (2) always distinct; (3) similar in 5-year-olds and distinct in 8-year-olds; or (4) distinct in 5-year-olds and similar in 8-year-olds (**Fig 1A**).

Pediatric neuroimaging comes with a number of challenges. For example, performance typically increases with age, such that tasks that require an active behavioral response confound developmental changes in activity with differences in performance [20]. Young children also struggle to lay still in a scanner for long periods of time, increasing motion-related noise in the data [21]. Therefore, even-related designs that estimate brain activity based on a limited number of trials may lack power and reliability [22]. To circumvent these issues, we used a block-design adaptation paradigm in which children were passively presented with blocks of nonsymbolic quantities (dot arrays) and Arabic numerals (digits) that were either similar (adaptation) or different (no-adaptation) (**Fig 1B**). Because the repeated presentation of a given stimulus leads to a decrease in activity in the region that processes that stimulus [23], comparing no-adaptation to adaptation blocks captures a neural adaptation effect in task-relevant regions [15,24].

Hypotheses in **Fig 1A** were tested using multivariate searchlight decoding across the whole brain [25], within but also between formats (**Fig 1C**). Within-format decoding involved (i) training cross-participants decoders to classify between adaptation and no-adaptation blocks of either dots or digits and (ii) testing the accuracy of decoders on different participants presented with stimuli in the same format. This was done both within the same age group but also

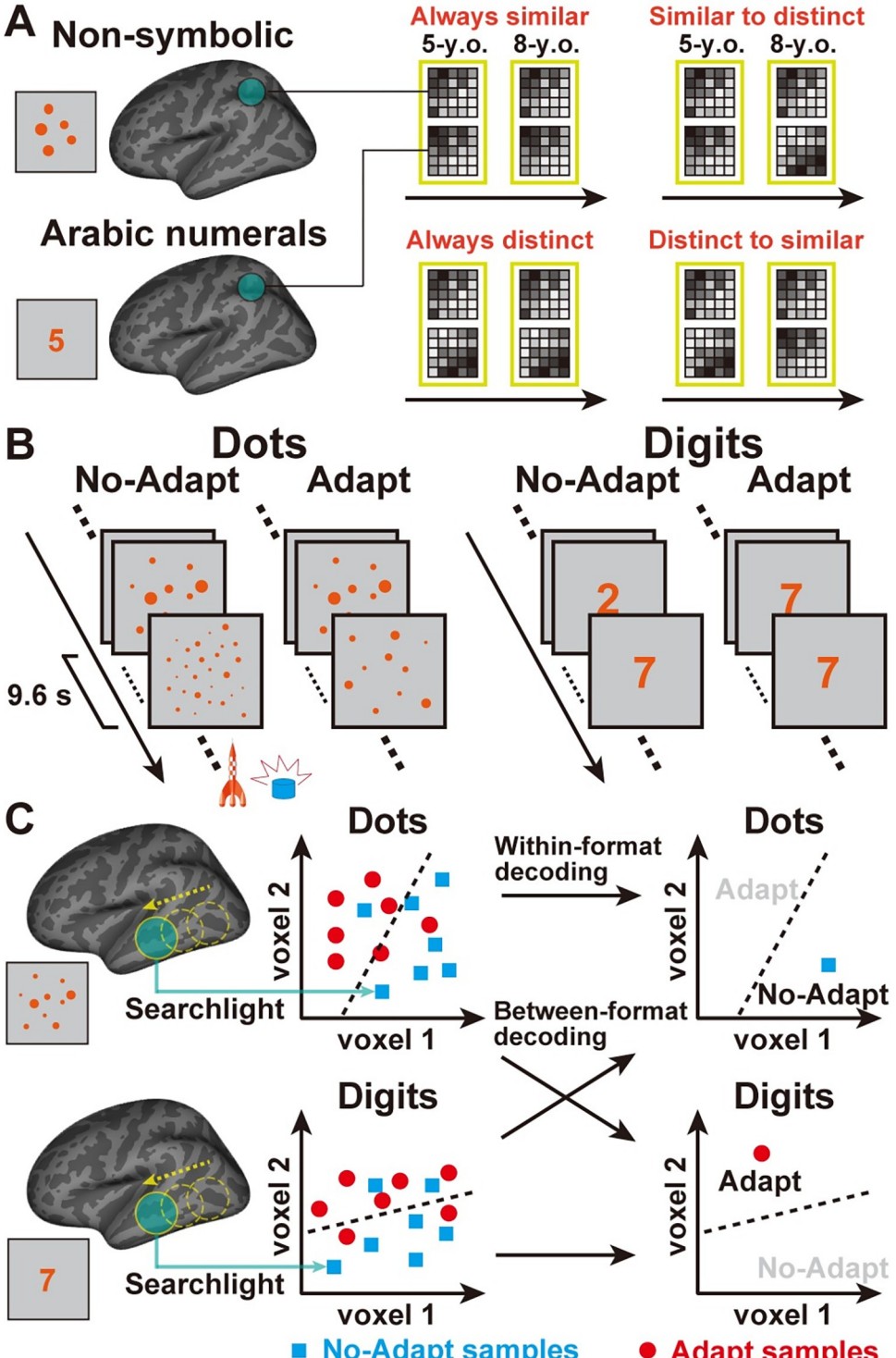

**Fig 1. Experimental design and analyses.** (**A**) Hypotheses regarding the similarity between neural representations of Arabic numerals and nonsymbolic quantities and how it might change with age. (**B**) Dots and digits were passively presented in adaptation and no-adaptation blocks. A target detection task was embedded in the adaptation task to ensure that participants were paying attention. (**C**) A pairwise classifier distinguishing adaptation from no-adaptation blocks of either dots or digits was constructed in each target searchlight sphere. It was subsequently applied to the left-out test samples, which were either of the same format (within-format decoding) or of a different format (between-format decoding).

across age groups to evaluate whether neural representations are stable through the first four years of formal education. Finally, the hypothesis that the relation between symbolic and non-symbolic representations might change with age was evaluated using between-format decoding. Specifically, within each age group ($n$ = 43 for 5-year-olds; $n$ = 46 for 8-year-olds), we tested whether decoders could accurately classify between adaptation and no-adaptation blocks of stimuli presented in a format that differed from the format they were trained on (e.g., dots to digits or digits to dots). This allowed us to evaluate whether any brain regions may represent quantities in an abstract (i.e., format-independent) manner and whether this may change with age.

## Results

### Child measures

To characterize the samples, children's intelligence quotient (IQ) and mathematical skills were assessed outside of the scanner using standardized (age-normalized) batteries (see **Materials and methods**). Average IQ, as measured by the NEMI-2 [26], was in the normal range in each age group. Specifically, 5-year-olds had an average IQ of 109 (standard deviation [SD] = 15, range = 76 to 146), while 8-year-olds had an average IQ of 112 (SD = 11, range = 83 to 135). There was no difference in standardized IQ between the two groups (Wilcoxon rank-sum test, $p$ = 0.18).

Measures of mathematical skills included standardized assessments of symbolic and non-symbolic numerical competence from two age-appropriate batteries, the TEDI-MATH for 5-year-olds [27] and the ZAREKI-R for 8-year-olds [28] (see **Materials and methods**). For 5-year-olds, measures involved symbolic number comparison and nonsymbolic quantity estimation. For 8-year-olds, measures involved symbolic number comparison and nonsymbolic quantity comparison. For nonsymbolic competence, the average standardized score was 104 in 5-year-olds (SD = 5, range = 85 to 105) and 108 in 8-year-olds (SD = 18, range = 57 to 123). For symbolic competence, the average standardized score was 102 in 5-year-olds (SD = 13, range = 57 to 109) and 107 in 8-year-olds (SD = 7, range = 83 to 111). Therefore, although scores are difficult to compare between groups due to the use of different batteries, average scores of 5- and 8-year-olds were in line with typical age expectation. This also indicates that raw performance was higher for 8- than 5-year-old.

### Within-format searchlight decoding

Based on multivariate activity from each target searchlight sphere, we trained cross-participant decoders to classify between adaptation and no-adaptation blocks of either dots or digits within each age group. Using leave-one-out cross-validation (LOOCV), we then tested whether the decoders could accurately classify (i.e., above chance level) between adaptation and no-adaptation blocks of left-out test samples from the same format (i.e., within-format decoding). Within each age group, we found several regions in which adaptation and no-adaptation blocks of dots, as well as adaptation and no-adaptation blocks of digits, could be accurately classified. These included regions of the occipital, frontal, and parietal cortices, including the IPS (**Fig 2**). **S1 and S2 Figs** show the unthresholded within-format accuracy maps for 5-year-olds (Dots, **S1A Fig**; Digits, **S1B Fig**) and 8-year-olds (Dots, **S2A Fig**; Digits, **S2B Fig**).

As can be seen in **Fig 2**, there appears to be differences but also similarities in the neural representations of symbolic and nonsymbolic quantity between groups. First, differences between groups were formally assessed by comparing accuracy maps using a whole-brain two-sample permutation test. Although there was no brain region in which either dot or digit decoding accuracy was larger in 5-year-olds than in 8-year-olds, decoding accuracy was larger

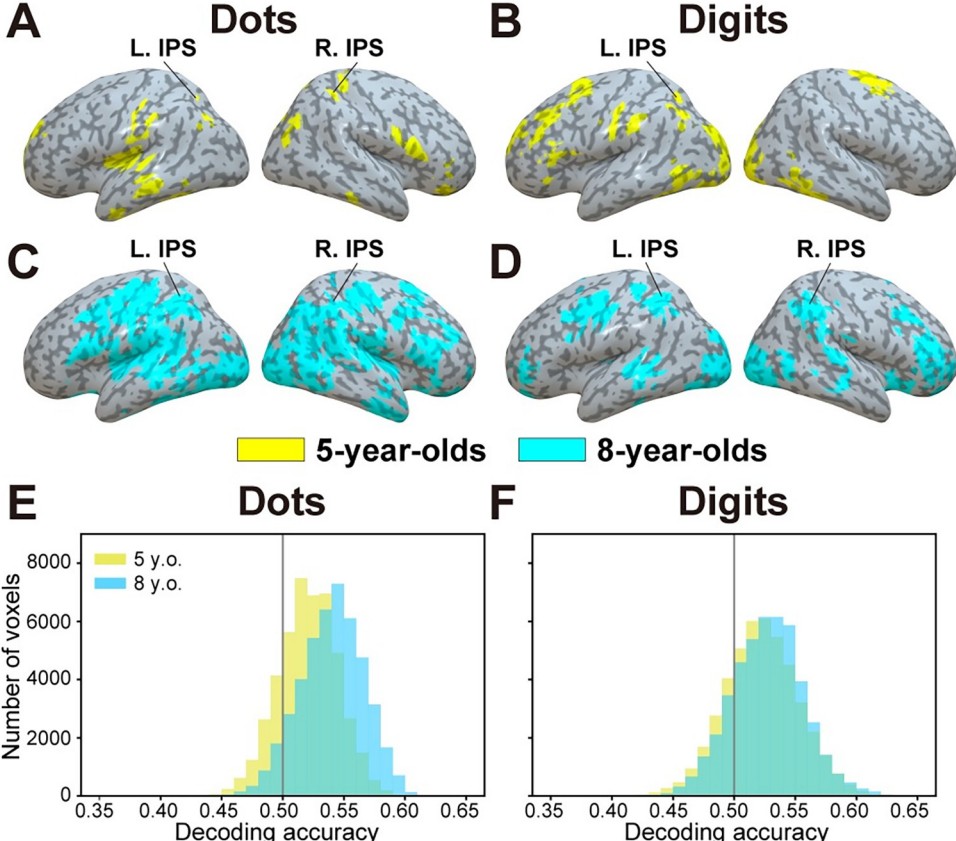

**Fig 2. Within-format decoding for each age group.** (**A-B**) For 5-year-olds, brain regions in which activity could accurately classify between adaptation versus no-adaptation blocks of (**A**) dots and (**B**) digits based on training with the same format. (**C-D**) For 8-year-olds, brain regions in which activity could accurately classify between adaptation versus no-adaptation blocks of (**C**) dots and (**D**) digits based on training with the same format. Only statistically significant clusters are shown (sign permutation test, voxel-level $p < 0.005$, cluster-level $p < 0.05$ with false discovery rate correction). IPS, intraparietal sulcus; L, left hemisphere; R, right hemisphere. (**E-F**) Histogram of mean decoding accuracy across all cortical voxels for (**E**) dots and (**F**) digits, plotted for both 5- and 8-year-olds. The underlying data can be found online (see Materials and methods). Specifically, data supporting panels (**A-D**) can be found in files "RawDecAcc_LOOCV_[5yo/8yo]_[Dots/Digits]_[Subjects' ID].nii" in the "RawDecAcc" folder, while data supporting panels (**E-F**) can be found in files "HistPlot_[Dots/Digits]_[5yo/8yo]_SourceData.npy" in the "GraphSourceData" folder.

in 8-year-olds than in 5-year-olds in several brain regions (**Fig 3A and 3B** and **S1** and **S2 Tables**). These notably included the IPS and regions of the occipital cortex for dots, as well as regions of the prefrontal cortex for digits. Higher within-format decoding accuracy in 8-year-olds can also be seen in histograms across all cortical voxels (Wilcoxon signed-rank test, $p < 0.001$; **Fig 2E and 2F**).

Second, we evaluated commonalities in neural representations between groups by (i) training decoders to classify between adaptation and no-adaptation blocks in a given age group (i.e., 5-year-olds or 8-year-olds) and (ii) testing whether decoders could accurately classify between adaptation and no-adaptation blocks of the same format but in the other age group (e.g., 8-year-olds or 5-year-olds). For each format, accuracy maps resulting from both analyses (i.e., one using 5-year-olds as a training set and the other using 8-year-olds) were combined in a conjunction analysis to identify the regions in which neural representations were most stable between groups (**Fig 3C and 3D** and **S3** and **S4 Tables**; see **S3 Fig** for accuracy maps associated with each direction-specific analysis). **S4 Fig** shows the unthresholded across-groups

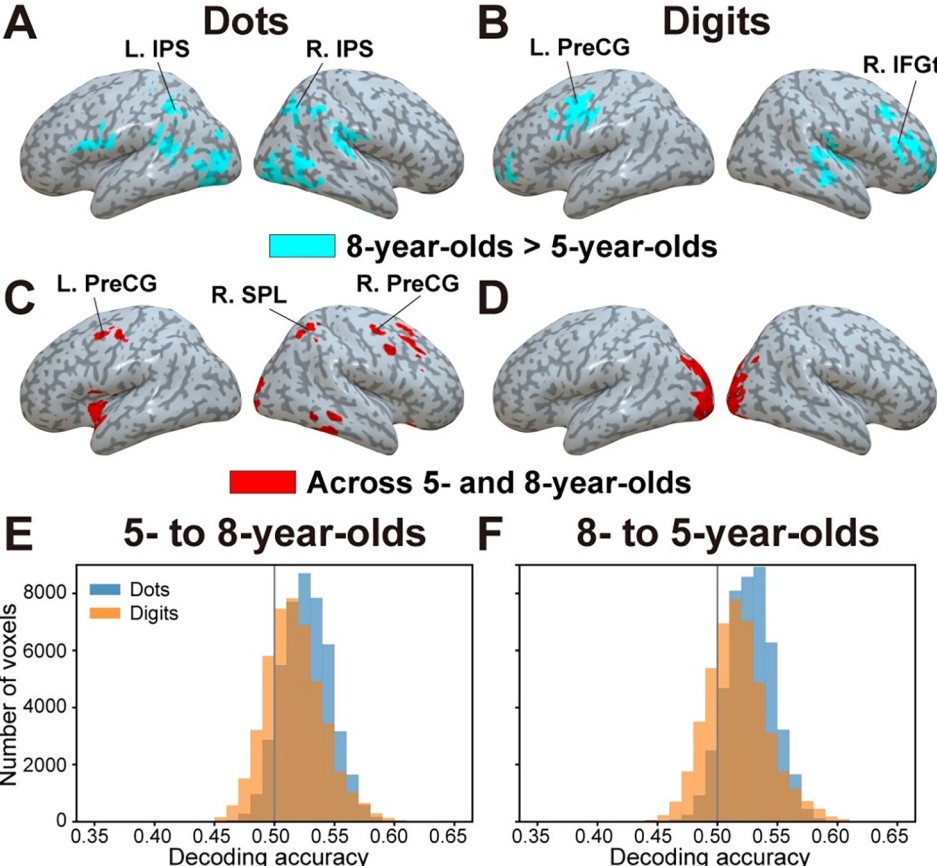

**Fig 3. Differences and similarities in within-format decoding across groups.** (**A-B**) Brain regions in which within-group decoding accuracy was larger in 8-year-olds than in 5-year-olds for (**A**) dots and (**B**) digits (based on training with the same format). (**C-D**) Brain regions in which activity could accurately classify between adaptation versus no-adaptation blocks of (**C**) dots and (**D**) digits in one group based on training with the same format in the other group (conjunction analysis). IFGtri, triangular part of inferior frontal gyrus; IPS, intraparietal sulcus; PreCG, precentral gyrus; SPL, superior parietal lobule. (**E-F**) Histogram of mean decoding accuracy across all cortical voxels for (**E**) 5- to 8-year-olds and (**F**) 8- to 5-year-olds directions, plotted for both dots and digits. The underlying data can be found online (see Materials and methods). Specifically, data supporting panels (**A-B**) can be found in files "RawDecAcc_LOOCV_[5yo/8yo]_[Dots/Digits]_[Subjects' ID].nii" in the "RawDecAcc" folder; data supporting panels (**C-D**) can be found in files "RawDecAcc_[5to8yo/8to5yo]_[Dots/Digits]_[Subjects' ID].nii" in the "RawDecAcc" folder; and data supporting panels (**E-F**) can be found in files "HistPlot_[Dots/Digits]_[5to8yo/8to5yo]_SourceData.npy" in the "GraphSourceData" folder.

accuracy maps dots (S4A Fig) and digits (S4B Fig). Results showed that adaptation versus no-adaptation blocks of dots could be accurately distinguished across groups (i.e., using one group as a training set and the other group as a testing set) in the bilateral precentral gyrus (PreCG) and right superior parietal lobule (SPL). This was only the case in the occipital cortex for adaptation versus no-adaptation blocks of digits. Higher decoding accuracy in dots can also be seen in histograms across all cortical voxels ($p < 0.001$; **Fig 3E and 3F**). Therefore, results of within-format decoding analyses revealed that the neural representations of Arabic numerals, but also of nonsymbolic quantity, clearly expanded from age 5 to age 8.

## Between-format searchlight decoding

We then examined whether the neural representations of symbolic and nonsymbolic quantity were similar within each group and whether this similarity depended on the age group. In

each age group, we tested whether decoders trained to classify between adaptation and no-adaptation blocks in a given format (i.e., dots or digits) could accurately classify between adaptation and no-adaptation blocks of left-out test samples from the other format (i.e., between-format decoding). Specifically, we reasoned that a brain region in which quantity is represented in a format-independent manner should show significant decoding accuracy both (i) when the classifier is trained with dots and tested on digits and (ii) when the classifier is trained with digits and tested on dots. For each age group, accuracy maps resulting from both analyses (i.e., one using dots as a training set and the other using digits) were combined in a conjunction analysis to identify the regions in which neural representations were format independent.

Although we did not find any region showing this format-independent pattern in 8-year-olds, a region of the right inferior parietal lobule (IPL) (and of the left PreCG) showed such a pattern in 5-year-olds (**Fig 4A** and **S5 Table**; see **S5 Fig** for accuracy maps associated with each direction-specific analysis). A direct comparison of accuracy maps between groups showed higher between-format decoding accuracy for 5-year-olds than for 8-year-olds in the right IPL, as well as in regions of the right middle frontal gyrus (MFG) and the triangular part of inferior frontal gyrus (IFGtri) (**Fig 4B** and **S6 Table**). No region showed higher between-format decoding accuracy for 8-year-olds than for 5-year-olds. Higher between-format decoding accuracy in 5-year-olds can also be seen in histograms across all cortical voxels ($p < 0.001$;

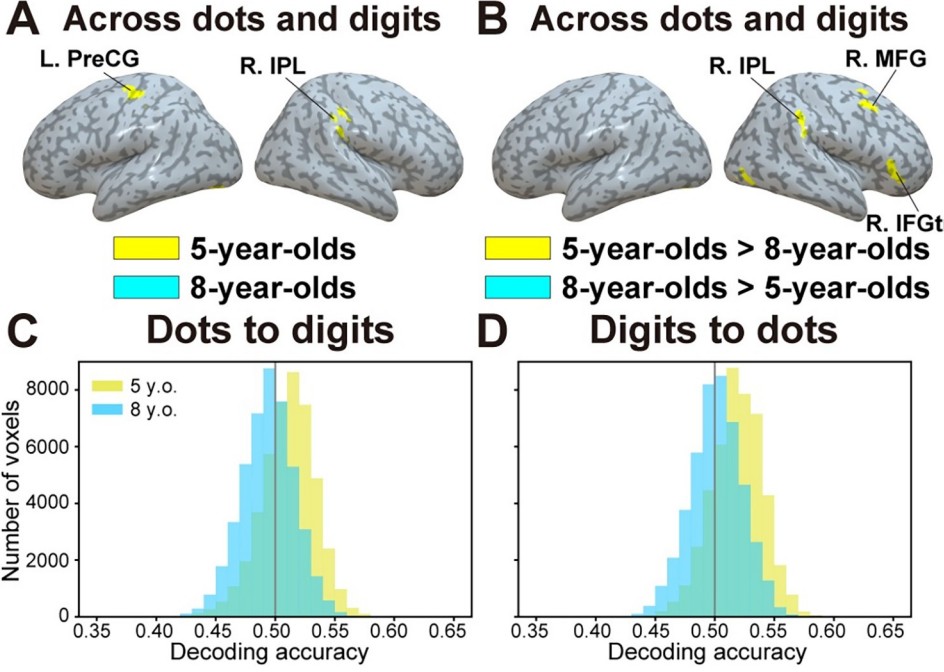

**Fig 4. Between-format decoding.** (**A**) Brain regions in which activity could accurately classify between adaptation versus no-adaptation blocks of quantity presented in one format (dots or digits) based on training with the other format (digits or dots) (conjunction analysis). (**B**) Brain regions in which between-format decoding accuracy was larger in 5-year-olds than in 8-year-olds. IFGtri, triangular part of inferior frontal gyrus; IPL, inferior parietal lobule; MFG, middle frontal gyrus; PreCG, precentral gyrus. (**C-D**) Histogram of mean decoding accuracy across all cortical voxels for (**C**) dots to digits and (**D**) digits to dots directions, plotted for both 5- and 8-year-olds. The underlying data can be found online (see Materials and methods). Specifically, data supporting panels (**A-B**) can be found in files "RawDecAcc_LOOCV_[5yo/8yo]_[Dots2Digits/Digits2Dots]_[Subjects' ID].nii" in the "RawDecAcc" folder, while data supporting panels (**C-D**) can be found in files "HistPlot_[Dots2Digits/Digits2Dots]_[5yo/8yo]_SourceData.npy" in the "GraphSourceData" folder.

**Fig 4C and 4D**). **S6 Fig** shows the unthresholded between-format accuracy maps for 5-year-olds (**S6A Fig**) and 8-year-olds (**S6B Fig**). Therefore, these results indicate some degree of similarity in the neural representations of symbolic and nonsymbolic quantity in 5-year-olds but not in 8-year-olds.

## Control analyses

The results above suggest the existence of a format-independent neural representation of quantity at age 5. It is important, however, to rule out four other potential explanations for this finding. First, it is possible that this result is not specific to the representation of quantity per se but instead reflects some domain-general similarities in the neural mechanisms associated with repetition-induced adaptation effects in young children. To examine this possibility, 5-year-olds were also presented with adaptation and no-adaptation blocks of letters (**Fig 5A**). Letters are not only perceptually similar to digits; they are also culturally invented symbols that young children learn early in school. Critically, however, letters do not carry any information about numerical quantity. We thus assessed between-format decoding accuracy across dots and letters. Specifically, we tested whether decoders trained to classify between adaptation and no-adaptation blocks of dots could accurately classify between adaptation and no-adaptation blocks of letters, as well as the other way around. Accuracy maps resulting from both analyses (i.e., one using dots as a training set and the other using letters) were then combined in a conjunction analysis. In contrast to the results obtained in the between-format decoding analysis across dots and digits (see above), between-format decoding accuracy across dots and letters was not higher than chance in any cortical region (see **S7 Table**). In fact, a whole-brain two-sample permutation test revealed that decoding accuracy between dots and digits was higher than decoding accuracy between dots and letters in the right IPL (**Fig 5B and 5C** and **S8 Table**). **S7 Fig** shows the unthresholded between-format accuracy maps across dots and letters. Additional analyses examining within-format decoding for letters and between-format decoding across digits and letters in 5-year-olds are presented in the Supporting information (**S8 Fig and S9 Table**).

Second, because stimuli were passively presented to participants, it is possible that the tasks might have differed in levels of attentional engagement within each age group. For example, a difference in attentional engagement between tasks might have jeopardized our ability to detect similarities between neural responses to adaptation (e.g., in 8-year-olds in which there was no between-format decoding). To examine this possibility, we asked participants to detect a randomly appearing target over the course of the experiment. On average, 5-year-olds detected 75% of targets (SD = 28) in the dot adaptation task, 79% of targets (SD = 25) in the digit adaptation task, and 80% of targets (SD = 22) in the letter adaptation task. Eight-year-olds detected 91% of targets (SD = 14) in the dot adaptation task and 92% of targets (SD = 15) in the digit adaptation task. Target detection rates of three tasks were largely above chance in each group (Wilcoxon signed-rank test, Bonferroni correction for multiple comparisons, adjusted $p < 0.001$), and there was no difference in target detection rate among the tasks within each age group (between the three tasks in 5-year-olds: Wilcoxon signed-rank test: $p > 0.23$; between dots and digits adaptation tasks in 8-year-olds: $p = 0.44$). Therefore, children paid attention to the stimuli in all tasks, and levels of attention did not differ between tasks.

Third, although 5-year-olds and 8-year-olds did not differ with respect to either in-scanner motion (see **Materials and methods**) or standardized IQ (see above), it remains possible that differences between groups in terms of motion or general cognitive functioning might have affected our results. To exclude these possibilities, we performed a multiple linear regression analysis (with intercept) using the six head motion parameters and IQ as independent

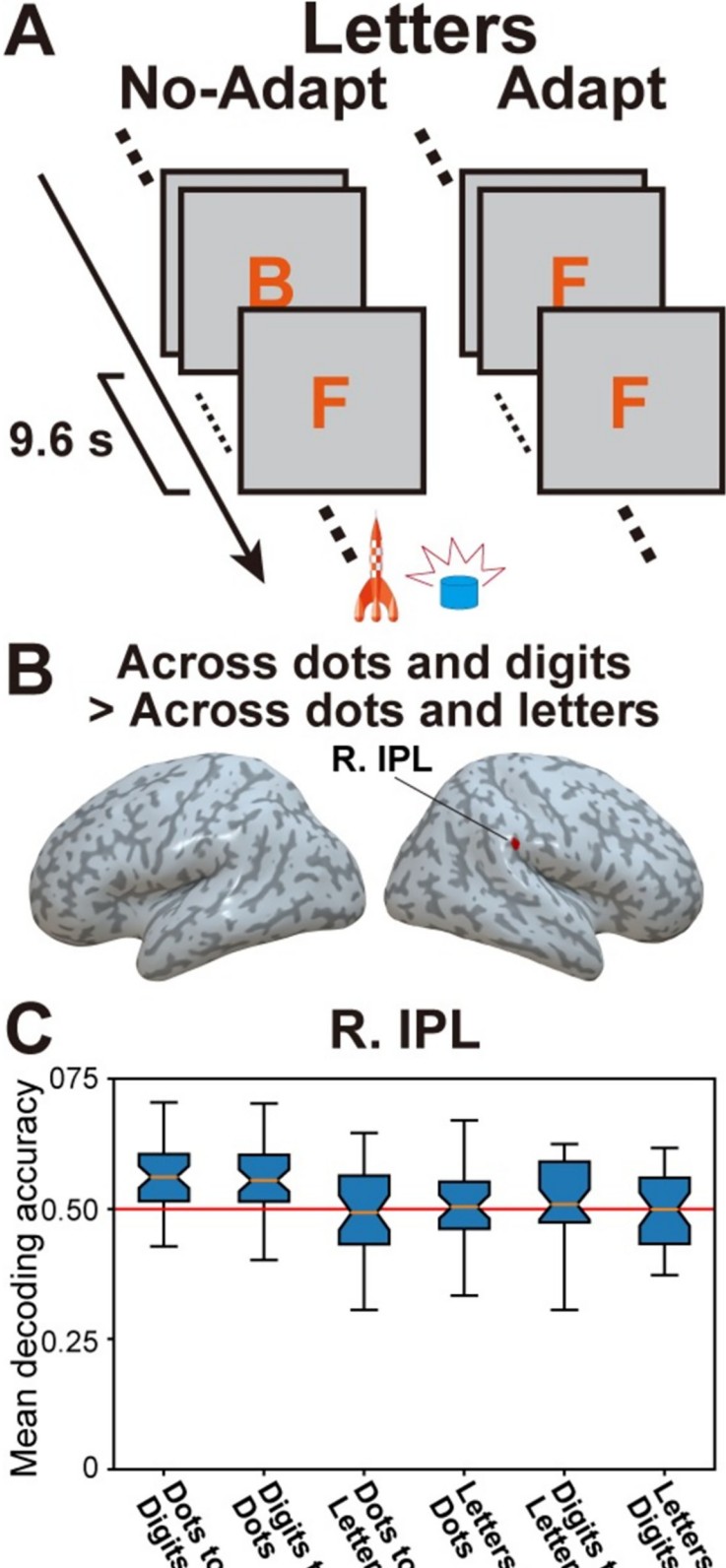

**Fig 5. Control analysis using the letter adaptation task.** (**A**) Letters were passively presented in adaptation and no-adaptation blocks. (**B**) Brain regions in which between-format decoding accuracy was larger across dots and digits compared to between-format decoding accuracy across dots and letters. (**C**) Mean decoding accuracy in the right IPL (defined in **Fig 4A**) for all decoding directions (dots to digits, digits to dots, dots to letters, letters to dots, digits to letters, and letters to digits). The underlying data can be found online (see Materials and methods). Specifically, data supporting panel (**B**) in this figure can be found in files "RawDecAcc_LOOCV_5yo_[Dots2Digits/Digits2Dots/Dots2Letters/Letters2Dots]_[Subjects' ID].nii" in the "RawDecAcc" folder, while data supporting panel (**C**) can be found in file "BoxPlot_SourceData.npy" in the "GraphSourceData" folder.

variables, as well as the decoding accuracy of each participant as a dependent variable. The predicted value associated with the independent variables was subtracted from the original decoding accuracy and was used as a new decoding accuracy. The group-level statistical significance was evaluated in the same manner as described above. Again, this analysis revealed similar results, with notably significant between-age decoding accuracy in the bilateral PreCG and higher between-format decoding accuracy for 5-year-olds in the right IPL (**S9 and S10** Figs). Therefore, differences between children's in-scanner motion or IQ did not appear to have influenced our results.

Finally, decoding accuracy might depend (to some extent) on the choice of analysis parameters, such as the type of cross-validation procedure and the voxel-level threshold that provides the basis for cluster-level correction [29]. To evaluate to what extent our results were sensitive to these choices, we performed additional analyses using a different type of cross-validation procedure and a different voxel-level threshold. First, we computed decoding accuracy maps using 10-fold cross-validation instead of LOOCV [30]. As shown in **S11 and S12** **Figs,** we obtained results that were very similar to our main findings. Notably, we found significant between-format decoding accuracy in the right IPL for 5-year-olds, but not for 8-year-olds. Second, we recalculated the significance of clusters in decoding accuracy maps using a voxel-level threshold of $p < 0.001$ instead of $p < 0.005$. Results were again very similar to our main findings. As shown in **S13 Fig**, we still found (1) greater within-format decoding accuracy in 8-year-olds than in 5-year-olds in frontal, parietal, and occipital brain regions; (2) significant within-format decoding accuracy across groups in the prefrontal and occipital cortex; and most importantly (3) significant between-format decoding in the right IPL for 5-year-olds, but not for 8-year-olds. Altogether, these results indicate that the present findings did not hinge upon a specific cross-validation method or a specific thresholding.

## Discussion

In the present cross-sectional study, we used fMRI to examine the emergence of the neural representations of Arabic numerals through early elementary education. Specifically, we performed a series of within- and between-format searchlight decoding analyses to investigate the relation between the neural representations of Arabic numerals and the neural representations of approximate nonsymbolic quantities from age 5 to age 8. Within-format decoding showed that Arabic numerals and nonsymbolic quantities were represented in distributed cortical regions in both 5-year-olds and 8-year-olds. While there was some similarity between age groups, there was also an expansion of the cortical territory dedicated to these representations. Between-format decoding demonstrated the existence of a format-independent representation of quantity in the right IPL. However, this format invariance was only observed in 5-year-olds, not in 8-year-olds. These results provide support for the symbolic estrangement hypothesis [16,19], which assumes that the relation between symbolic and nonsymbolic quantity weakens with exposure to formal math education.

## Cortical representations of symbolic and nonsymbolic quantity expand from age 5 to age 8

A number of previous studies have suggested that neural sensitivity to approximate nonsymbolic numerical quantity can be detected as early as in the first years of life, particularly in the right parietal cortex [1–4]. fMRI evidence has also demonstrated that changes of activity in the parietal and frontal cortices are associated with the development of numerical skills in children [31]. Our findings, however, are novel in that they suggest both stability and changes in the brain system underlying the representation of symbolic and nonsymbolic quantity in children as they are exposed to the first years of formal schooling.

On the one hand, we found that the representations of Arabic numerals and nonsymbolic quantities were represented in the bilateral occipital and parietal cortices of both 5- and 8-years-olds. These areas have long been the focus of previous studies investigating neural activity associated with symbolic and nonsymbolic quantity [6,9,13–16]. However, our results suggest the involvement of a more distributed brain system that also includes bilateral regions of the temporal and frontal cortices. Importantly, this finding was only made possible because we used searchlight decoding, allowing for the detection of multivariate representations across the whole brain. This notably contrasts with the majority of previous pediatric neuroimaging studies, which have often investigated whole-brain univariate activity [1–4,32]. Nonetheless, it is interesting to note that our study is not the first to show that the temporal and frontal cortices are involved in the processing of symbolic and nonsymbolic quantity in children. For example, Kovas and colleagues [33] and Kersey and Cantlon [4] found that a distributed system is associated with the processing of nonsymbolic numerosity in children from age 3 to age 10, including the parietal, occipital, temporal, and frontal cortices. Using a searchlight approach, Bulthé and colleagues [12] also reported that nonsymbolic quantity could be decoded across the whole brain. Interestingly, we found here some degree of stability in the neural representation of nonsymbolic quantity. Specifically, decoders trained with brain activity from 5-year-olds accurately classified between activity associated with adaptation versus no-adaptation blocks of dots in 8-year-olds and vice versa. This was notably the case in the right SPL and bilateral PreCG. However, no representational stability was found for the neural representation of Arabic numerals in the parietal, frontal, or temporal cortex between age 5 and age 8, most likely because symbolic numerical skills change massively during that time.

On the other hand, we found that the representations of symbolic and nonsymbolic quantity significantly expanded from age 5 to age 8. This was notably the case in the bilateral IPS and occipital cortex for dots, as well as in the prefrontal cortex for digits. Arguably, children are increasingly exposed to symbolic numbers from age 5 to age 8 (which corresponds to the first four years of formal education). This is likely to explain the expansion of territory dedicated to the representation of digits in the prefrontal cortex, as this region has often been found to be involved in symbolic quantity and arithmetic processing in adults [34]. Note, however, that there was also an expansion of the cortical regions representing nonsymbolic quantity, most notably around the IPS. Although nonsymbolic numerosity representations are evolutionarily old and therefore emerge early in children [1–4], this finding is consistent with the idea that the acquisition of symbolic number skills during the first four years of education may still refine the cognitive mechanisms supporting nonsymbolic quantity processing [35].

## Cortical representations of symbolic and nonsymbolic quantity become estranged from age 5 to age 8

Between-format decoding revealed that the first four years of formal education had a significant influence on the relation between the brain representations of symbolic and nonsymbolic

quantity. Specifically, although we found a region of the right IPL in which quantity was represented in a format-independent manner in 5-year-olds, this format independence was largely absent in 8-year-olds. Between-format decoding accuracy was also significantly higher in 5-year-olds than in 8-year-olds, indicating that the groups statistically differed when compared with each other. This age dependence of the relation between symbolic and nonsymbolic representations is in line with a recent study that reported a correlation between arithmetic skills and representational similarity across symbolic and nonsymbolic numerical processing in elementary school children but not in adolescents [36].

It has long been argued that the acquisition of symbolic numerical skills in humans builds on an evolutionary-old representation of approximate nonsymbolic quantity [5,37]. Neural evidence for this hypothesis comes from a limited number of studies in adults showing format invariance in the brain representation of symbolic and nonsymbolic quantity, particularly in the parietal cortex [7,9]. Yet, a growing number of recent studies, also in adults, have failed to find such format invariance. Some have thus claimed that the link between symbolic and nonsymbolic representations might be more tenuous than previously thought [12–15]. Critically, our findings suggest that the representation of nonsymbolic quantity may still scaffold the acquisition of symbolic numerical skills in young children. It is possible that the inconsistent results observed in the adult literature stem from the fact that adults have had years of experience manipulating numerical symbols, such that their symbolic numerical skills might be integrated into an independent symbolic system [19,38]. More generally, our findings are a reminder that it is difficult to infer the mechanisms underlying the acquisition of numerical skills from studies in adults.

### Overlap in the cortical representations of symbolic and nonsymbolic quantity in young children is specific to numerical information

Unlike most previous fMRI adaptation studies [3,9,32], our study focuses on multivariate activity patterns during adaptation periods rather than univariate activity associated with a deviant stimulus. Although this makes our design particularly robust and well suited to pediatric neuroimaging, a downside is that we could not evaluate whether decoding was influenced by numerical properties of the stimuli (e.g., distance between quantities). Therefore, it could be argued that the overlap between the neural representations of dots and digits in 5-year-olds is mainly due to domain-general adaptation effects. To rule out this possibility and assess the specificity of the overlap between symbolic and nonsymbolic quantity, we also presented participants with symbolic stimuli that did not involve any quantitative information, i.e., letters. Critically, we did not find significant between-format decoding between dots and letters anywhere in the cortex (and we found higher decoding accuracy between dots and digits than that between dots and letters in the right IPL). This indicated that the overlap between the neural representations of dots and digits in the IPL of 5-year-olds was specific to numbers (**Figs 4 and 5**).

### Cortical representations of digits and letters overlap in young children

Interestingly, the lack of overlap between representations of dots and letters was not due to a failure to find shared representations between letters and any other types of stimuli. Indeed, in 5-year-olds, there was significant between-format decoding across letters and digits in both the bilateral fusiform gyrus (FG) and the right PreCG (**S8 Fig**). This suggests that both of these regions might be involved in representing symbolic information more generally. Consistent with this idea, several studies notably suggest that the FG might include areas dedicated to the identification of both words and numerals from low-level visual features [39–42]. It is also possible that the neural overlap between digits and letters in the right PreCG is due to a common representation of stimuli features, independently of quantity (e.g., visual features).

## Limitations

To our knowledge, our study is unique in its investigation of the neural representations of symbolic and nonsymbolic quantity using multivariate decoding in a large sample of children. However, it is also worth considering a number of potential limitations. First, although we interpret differences between 5- and 8-year-olds in terms of changes associated with learning and development, it is important to note that our design is cross-sectional. In other words, it is possible that the sample of 5-year-olds and the sample of 8-year-olds differ in a number of ways, which might make them not entirely comparable. In our view, this possibility is mitigated by the fact that we acquired a comprehensive range of background measures in each group, ensuring that they are comparable in terms of age-normalized IQ and math skills, as well as in terms of socioeconomic status (SES) (see **Materials and methods**). In any case, our findings should provide the foundation for future studies investigating longitudinal changes in neural representations within the same participants.

Second, it is always important to consider whether our participants are representative of the general population. To ensure the representativity of our samples, we recruited children from various areas within the Lyon metropolitan area. We were also inclusive of children with relatively low cognitive functioning, only excluding children with an IQ lower than the 2.5th percentile because they would not have been able to adequately complete the tasks. Overall, the mean IQ in both groups can be considered high average, with a score of 109 for 5-year-olds and 112 for 8-year-olds. Such high average numbers are relatively common in the developmental cognitive literature [32,43]. They might be inevitable considering (1) the challenges raised by recruiting diverse samples in lab-based developmental neuroimaging studies and (2) the need to remove from the fMRI data analyses participants with high motion and poor task performance (both of these being potentially related to levels of attention and general cognitive functioning). In our view, the most critical aspect for generalizability is to ensure that none of the findings are related to individual differences in cognitive function, which we verified in the present study (see control analyses above). Therefore, although future studies might extend our findings to children with lower cognitive functioning, we believe that our findings remain generalizable to the population.

Third, studying task-related changes in neural processing across development is always challenging because such changes are often confounded with differences in behavioral performance [20]. For this reason, we used adaptation tasks that do not require any overt behavior. As a result, we largely interpret differences between age groups as coming from differences in the passive processing of symbolic and nonsymbolic quantity. However, it could be argued that such differences might also be (at least partly) driven by differences in attentional levels between groups. We think that this is unlikely because analysis of the concurrent target detection task showed that even 5-year-olds were paying significant attention to the stimuli. It is also interesting to note that if attention was driving the between-group differences in between-format decoding, such a finding would have been more likely in 8-year-olds than in 5-year-olds (as levels of attention are expected to increase with age). Yet, we found larger between-format decoding in 5-year-olds than in 8-year-olds. Nonetheless, we acknowledge that even passively processing symbolic and nonsymbolic quantity might involve a number of cognitive processes that are difficult to control for and might contribute to the differences observed (e.g., working memory, metacognition).

Fourth, although passive adaptation tasks are well suited to examine differences in brain activity between children of different ages, a drawback is that they do not make it possible to collect indicators of task performance. Therefore, our results do not allow us to make any conclusion regarding how well the participants could identify and discriminate symbolic and

nonsymbolic quantities at the behavioral level. However, there is overwhelming evidence that performance on symbolic and nonsymbolic quantity tasks increases between the ages of 5 and 8 [44,45], which is also suggested by the standardized measures collected outside of the scanner in the present study. Of course, this does not mean that there are no individual differences in performance within each age group. Even though assessing the relation between those individual differences and decoding performance would be informative, such analyses would require much larger sample sizes than those used in the present investigation [46].

Finally, the main goal of the present study was to test the influential hypothesis that learning symbolic numbers builds on an approximate sense of quantity that is associated with the ANS [5]. To elicit ANS processing, we adapted our nonsymbolic stimuli from previous tasks in which participants are typically presented with relatively large sets of dots that cannot be enumerated (thereby preventing exact symbolic processing). However, because young children in kindergarten do not master the symbolic place-value notation, our symbolic stimuli did not include double-digit numbers and were restricted to Arabic numerals. In other words, symbolic and nonsymbolic stimuli did not represent the same quantities. Therefore, although our results inform about the relation between the neural representations of Arabic numerals and approximate nonsymbolic quantities in young children, our conclusions might not extend to the relation between exact symbolic and exact nonsymbolic quantities.

## Conclusions

In sum, our findings indicate that the brain representations of Arabic numerals and approximate nonsymbolic quantities expand during the first years of elementary education. Furthermore, we show some overlap in the neural representations of numerals and approximate nonsymbolic quantities in the parietal cortex of 5-years-olds. However, this overlap disappeared in 8-year-olds, suggesting that the representation of symbolic quantity becomes independent from that of approximate nonsymbolic quantity in the first years of formal math education. Overall, these results remain consistent with the cultural recycling hypothesis, which assumes that a culturally developed system of symbolic quantity is grounded in an evolutionarily older system for approximate nonsymbolic quantity representation [37,47]. However, they also suggest that the brain representations of symbolic and approximate nonsymbolic quantity become estranged with learning and enculturation.

## Materials and methods

### Participants

Two hundred and six children (133 children of approximately 5 years of age and 73 children of approximately 8 years of age) were recruited to participate in the experiment, which consisted of a first behavioral and a second fMRI session. Children were recruited through flyers sent to schools and advertisements on social media. All children were native French speakers. Parents gave written informed consent, and children gave their assent to participate in the study. The study was approved by a French ethics committee (Comité de Protection des Personnes Sud-Est 2 and Comité pour la Protection des Personnes Sud-Ouest et Outre-Mer). Families were paid 80 euros for their participation. During the first session, all participants came to the lab to be familiarized with the fMRI environment in a mock scanner and to complete psychometric testing. Forty-four participants did not continue to the fMRI session, either because children were not comfortable with the fMRI session (as determined by mock scanning) or because they met exclusion criteria. Specifically, children were not invited to the fMRI session if they had regular visits to a speech-language pathologist ($n = 3$), an IQ lower than the 2.5th percentile ($n = 3$), delayed speech and language acquisition (n = 1), diagnosis of attention

deficit disorder (n = 1), and incomplete behavioral testing (n = 2). Out of the 162 children who participated in the fMRI session, 49 children were not able to complete at least one run of dot adaptation and one run of digit adaptation. Twenty-three children were also excluded for excessive motion in the scanner (see criteria below). Finally, one participant was excluded because of incomplete fMRI acquisition.

Therefore, our main fMRI sample consisted of 89 children who had at least one run of data analyzable in both the digit and dot adaptation tasks. Forty-three children were in the 5-year-old group (mean age, 5.41, SD = 0.43; 18 females), while 46 children (age, mean = 8.49, SD = 0.36; 15 females) were in the 8-year-old group. Because neural adaptation to letters was also measured in 5-year-old children, additional control analyses include a subgroup of children who had at least one run of letter adaptation in addition to one run of dot adaptation and one run of digit adaptation. This control analysis included 38 children in the 5-year-old group. Previous studies explored the relation between brain activity and the home learning environment in some of these participants [24,48].

Family SES of children was assessed using the income of the parent who was with the child during testing. Parental income ranged from 500 to 5,500 euros (mean = 1,893, SD = 1,164) across all families included in the main analyses. Given that the median monthly income in France is about 1,700 euros [49], family SES ranged from low to high. There was no difference between children who were included in the final analyses and those who were excluded with respect to parental income (stats).

## Child measures

We assessed children's IQ and numerical skills using age-appropriate tests outside of the scanner. IQ was estimated using the NEMI-2 standardized intelligence test [26]. The test uses measures of verbal intelligence and matrix reasoning to provide a standardized score of full-scale IQ. Symbolic and nonsymbolic numerical skills were estimated using subsets of TEDI-MATH [27] for 5-year-olds and subtests of the ZAREKI-R for 8-year-olds [28]. TEDI-MATH subtests were the "Estimation visuelle de quantités" and "Comparaison de deux nombres écrits." While the former subtest requires children to estimate the number of objects presented within two or five seconds, the latter requires children to decide which of two written symbolic numbers is the largest. ZAREKI-R subtests included the "Comparaison de patterns de points dispersés" and "Comparaison de nombres arabes." While the former subtest requires children to decide which of two dot patterns included the largest number of dots within 1 second, the latter requires children to decide which of two written symbolic numbers is the largest.

## Adaptation task

We adapted from [32] a task in which a series of stimuli were passively presented in blocks at the center of the screen, either in adaptation or no-adaptation conditions. This task differs from typical adaptation designs used in the numerical cognition literature in that we focused here on the decrease of activity associated with the adaptation period instead of the "rebound" activity that would be associated with a deviant presented after this adaptation period [9]. Although adaptation-related decreases in activity and rebound activity capture similar processes (including activity associated with numerosity processing in the IPS; [4]), focusing on the adaptation period involves block designs that induce a larger adaptation effect than an event-related design [50]. This is particularly desirable when conducting relatively short experiments with young children. The scanning involved four versions of that task, with different types of stimuli: dots, digits, letters, and words. The word adaptation task was not examined here (see [48] for an analysis of that task). In the dot adaptation task, stimuli were dot arrays

(ranging from 6 to 60 dots), i.e., set sizes in the dot adaptation task exceeded the subitizing range and the counting range (for the most part) [17]. Adaptation blocks consisted of the repetition of the same number of dots eight times, while no-adaptation blocks consisted of the presentation of eight different numbers of dots (the number of dots in an array differed from the previous array by a ratio from 1:2 to 1:8). To control for nonnumerical parameters, arrays systematically varied in terms of convex hull (i.e., smallest contour around the array of dots), aggregate surface of the dots, density (i.e., aggregate surface divided by the convex hull), average diameter, and contour length. In the digit adaptation task, stimuli were digits ranging from 1 to 8, which are known to children as young as 5 (which is not necessarily the case for double-digit numbers). In the letter adaptation task, stimuli were letters of the alphabet ("A," "B," "C," "D," "E," "F," "M," "R," and "S") presented in capital. In the digit and letter adaptation tasks, adaptation blocks consisted of the same digit or letter that was repeatedly presented eight times, while no-adaptation blocks consisted of the presentation of eight different digits or letters. Each task was always presented in different functional runs. The letter adaptation task was only presented to 5-year-olds and served as a control task in the present study.

## Experimental timeline

The experimental timeline was the same for all tasks and was also identical to [32]. The task was presented using Psychopy [51]. In each block, stimuli remained on the screen for 700 ms, with a 500-ms interstimulus interval (for a total block duration of 9.6 seconds). Ten adaptation blocks and ten no-adaptation blocks were presented along with ten blocks of visual fixation (duration = 9.6 seconds) in each run. Block presentation was pseudorandomized such that two blocks of the same type could not follow each other. To ensure that participants paid attention to the task in the scanner, ten target stimuli (a picture of a rocket) randomly appeared in each run (outside of blocks). Participants were asked to press a button every time this target appeared. Note that the behavioral data of four participants in the 5-year-old group could not be collected because of a technical issue.

## fMRI data acquisition

Images were collected using a Siemens Prisma 3 T MRI scanner with a 64-channel receiver head–neck coil (Siemens Healthcare, Erlangen, Germany) at the CERMEP Imagerie du vivant in Lyon, France. The blood oxygenation level-dependent (BOLD) signal was measured with a susceptibility-weighted single-shot echo planar imaging sequence. Imaging parameters were as follows: repetition time (TR) = 2,000 ms, echo time (TE) = 24 ms, flip angle = 80˚, field of view (FOV) = 220 × 206 mm$^2$, resolution = 1.72 × 1.72 mm$^2$, slice thickness = 3 mm (0.48 mm gap), number of slices = 32. A high-resolution T1-weighted whole-brain anatomical volume was also collected for each participant. Parameters were as follows: TR = 2,400 ms, TE = 2.81 ms, flip angle = 8˚, FOV = 224 × 256 mm$^2$, resolution = 1.0 × 1.0 mm$^2$, slice thickness = 1.0 mm, number of slices = 192.

## fMRI data preprocessing

Images were preprocessed with SPM12 (Wellcome Department of Cognitive Neurology, London, UK). The first four images of each run were discarded to allow for T1 equilibration effects. Functional images were corrected for slice acquisition delays and spatially realigned to the first image of the first run to correct head movements. Using ArtRepair (https://github.com/BBL-lab/BBL-batch-system/tree/main/dependencies/Artrepair)) [52], functional volumes with a global mean intensity greater than 3 SDs from the average of the run or a volume-to-volume motion greater than 2 mm were identified as outliers and substituted by the

interpolation of the two nearest nonrepaired volumes. Participants with outliers in more than 20% of volumes were excluded from the analyses (5-year-olds, $n = 21$; 8-year-olds, $n = 2$). After outlier exclusion, the movement range was on average 0.06 (SD = 0.06), 0.16 (SD = 0.14), and 0.26 (SD = 0.22) mm in the x, y, and z direction, with 0.39 (SD = 0.50), 0.14 (SD = 0.13), and 0.09 (SD = 0.11) degrees of roll, pitch, and yaw for 5-year-olds. The movement range was on average 0.09 (SD = 0.09), 0.17 (SD = 0.15), and 0.35 (SD = 0.30) mm in the x, y, and z direction, with 0.50 (SD = 0.47), 0.21 (SD = 0.20), and 0.12 (SD = 0.14) degrees of roll, pitch, and yaw for 8-year-olds. There was no significant difference in head motion between 5- and 8-year-old children (Wilcoxon rank-sum test, Bonferroni correction for multiple comparisons, adjusted $p = 0.25$, 1.00, and 0.25 for x, y, and z translations, respectively; adjusted $p = 0.40$, 0.10, and 0.40 for pitch, yaw, and roll rotations, respectively).

A standard practice in developmental cognitive neuroscience is to normalize functional images into the same adult stereotaxic space [4,10,32]. However, it is worth considering the possibility that anatomical differences between children and adults might affect the quality of the normalization. Studies have found that anatomical differences between children from 5 to 8 and adults are small enough that they are beyond the resolution of fMRI experiments [53,54]. Therefore, considering the age of our participants and the resolution of our data, we normalized all individual brains into the standard adult Montreal Neurological Institute (MNI) space. This was done in two steps. First, after coregistration with the functional data, the structural image was segmented into gray matter, white matter, and cerebrospinal fluid by using a unified segmentation algorithm [55]. Second, the functional data were normalized to the MNI space by using the normalization parameters estimated during unified segmentation (normalized voxel size, $2 \times 2 \times 2$ mm$^3$).

Finally, low-frequency drift was removed using a median filter with a 120-second window. The response at each voxel was then standardized by subtracting the mean response and scaling it to the unit variance. In order to fully include brain signal changes associated with the adaptation effect, we used block averages as classification samples [56]. In particular, we averaged five TRs for each block (duration = 9.6 seconds), with a 6-second temporal delay associated with the BOLD signals. We thus obtained ten samples for both the adaptation and no-adaptation blocks from each participant.

## Searchlight decoding analysis

Whole-brain searchlight decoding analysis was performed using in-house python scripts, Nilearn [57], and Scikit-learn libraries [58]. For each cortical voxel (sphere center), average brain activity was extracted from all voxels located in the gray matter within a 5-voxel radius sphere. The gray matter mask was created using the WFU PickAtlas toolbox [59]. For each searchlight sphere, a cross-participant classifier (distinguishing adaptation from non-adaptation blocks) was constructed using a support vector machine with a radial basis function kernel and a regularization parameter C = 1.

We performed both within- and between-format searchlight decoding analyses. Within-format decoding involved training cross-participant decoders in classifying between adaptation and no-adaptation blocks and testing the accuracy of decoders on different participants presented with stimuli in the same format. We performed two within-format decoding analyses. First, we focused on participants within their age groups, assessing decoding accuracy using LOOCV. Specifically, within each age group, a classifier was trained using (N—1) participants' data from one of two formats (dots or digits) and tested with the left-out participant's data from the same format (N, total number of participants in the target age group). Second, we evaluated the stability of neural representations across age groups. This was done by (1)

using all participants' data as training samples of one of the two age groups (5- or 8-year-olds) and (2) testing accuracy with each participant in the other age group.

Between-format decoding was performed on participants within their age groups. Specifically, we tested whether decoders could accurately classify between adaptation and no-adaptation blocks of stimuli presented in a format that differed from the format they were trained on (e.g., dots to digits or digits to dots). Using LOOCV, a classifier was trained using (N—1) participants' data from one of the three formats (dots, digits, or letters for 5-year-olds; dots or digits for 8-year-olds) and tested with the left-out participant's data from a format that was different from that used in the training phase.

Statistical significance was assessed using cluster-wise inference via a nonparametric sign permutation test [60,61]. Unlike parametric tests, which assume constant spatial smoothness across the brain, nonparametric tests do not make that assumption and are therefore much less sensitive to the cluster-defining threshold [62]. For each participant, the chance level accuracy (0.5) was subtracted from the decoding accuracy, and the sign of the resulting difference was randomly inverted. This procedure was repeated 50,000 times, and a null distribution of mean decoding accuracy (or difference between two mean decoding accuracies in case of direct comparison) was obtained in each voxel. Voxel-wise $p$-values were calculated as a number of randomly generated mean decoding accuracies larger than the actual mean decoding accuracy, divided by the repetition number (i.e., 50,000). Cluster-wise $p$-values were calculated for each target actual cluster, based on the number of randomly generated clusters in which the cluster size was larger than the target cluster size, divided by the total number of randomly generated clusters. In line with our most recent fMRI study using multivariate analyses [15], the statistical threshold was set at $p < 0.005$ for the voxel level and at $p < 0.05$ for the cluster level. Additional control analyses, however, use a voxel level of $p < 0.001$ to ensure that this parameter does not critically affect our conclusions (**S13 Fig**). The cluster-wise $p$-values were corrected for multiple comparisons using a false discovery rate procedure [63]. To evaluate brain regions in which neural representations were format invariant, we also performed a conjunction analysis. Based on the concept of minimum statistics [64], we calculated the maximum $p$-value of two decoding directions (e.g., dots to digits and digits to dots) in each voxel. The statistical significance was evaluated as in the normal decoding analysis. Anatomical labels were determined based on the automated anatomical labeling (AAL) atlas [65].

## Supporting information

**S1 Fig. Unthresholded maps of within-format decoding for dots. (A-B)** Voxels in which activity could accurately classify between adaptation versus no-adaptation blocks of dots based on training with the same format, shown with unthresholded maps for (**A**) 5-year-olds and (**B**) 8-year-olds. The underlying data supporting this figure can be found online in file "Dots_[5yo/8yo]_LogPval.nii" in "UnthresholdData" folder.
(TIF)

**S2 Fig. Unthresholded maps of within-format decoding for digits. (A-B)** Voxels in which activity could accurately classify between adaptation versus no-adaptation blocks of digits based on training with the same format, shown with unthresholded maps for (**A**) 5-year-olds and (**B**) 8-year-olds. The underlying data supporting this figure can be found online in files "Digits_[5yo/8yo]_LogPval.nii" in "UnthresholdData" folder.
(TIF)

**S3 Fig. Individual maps of similarities in within-format decoding across groups. (A-B)** Brain regions in which activity could accurately classify between adaptation versus no-

adaptation blocks of (**A**) dots and (**B**) digits in 8-year-olds based on training with the same format in the 5-year-olds. (**C-D**) Brain regions in which activity could accurately classify between adaptation versus no-adaptation blocks of (**C**) dots and (**D**) digits in 5-year-olds based on training with the same format in the 8-year-olds. Only statistically significant clusters are shown (sign permutation test, voxel-level $p < 0.005$, cluster-level $p < 0.05$ with false discovery rate correction). PreCG, precentral gyrus; SPL, superior parietal lobule. The underlying data supporting this figure can be found online in files "RawDecAcc_LOOCV_[5to8yo/8to5yo]_[Dots/Digits]_[Subjects' ID].nii" in "RawDecAcc" folder and in files "[Dots/Digits]_[5to8/8to5]_LogPval.nii" in "UnthresholdData" folder.
(TIF)

**S4 Fig. Unthresholded maps of within-format decoding across groups. (A-B)** Voxels in which activity could accurately classify between adaptation versus no-adaptation blocks of (**A**) dots and (**B**) digits in one group based on training with the same format in the other group (conjunction analysis), shown with unthresholded maps. The underlying data supporting this figure can be found online in files "[Dots/Digits]_5to8&8to5_LogPval.nii" in "Unthreshold-Data" folder.
(TIF)

**S5 Fig. Individual maps of between-format decoding. (A)** Brain regions in which activity could accurately classify between adaptation versus no-adaptation blocks of digits based on training with dots. (**B**) Brain regions in which activity could accurately classify between adaptation versus no-adaptation blocks of dots based on training with digits. (**C**) Brain regions in which between-format decoding accuracy was larger in 5-year-olds than in 8-year-olds, tested with digits based on training with dots. (**D**) Brain regions in which between-format decoding accuracy was larger in 5-year-olds than in 8-year-olds, tested with dots based on training with digits. IFGtri, triangular part of inferior frontal gyrus; IPL, inferior parietal lobule; MFG, middle frontal gyrus; PreCG, precentral gyrus. The underlying data supporting this figure can be found online in files "RawDecAcc_LOOCV_[5yo/8yo]_[Dots2Digits/Digits2Dots]_[Subjects' ID].nii" in "RawDecAcc" folder and in files "[Dots2Digits/Digits2Dots]_[5yo/8yo/5yo-8yo/8yo-5yo]_LogPval.nii" in "UnthresholdData" folder.
(TIF)

**S6 Fig. Unthresholded maps of between-format decoding. (A-B)** Voxels in which activity could accurately classify between adaptation versus no-adaptation blocks of quantity in one format (dots or digits) based on training with the other format (digits or dots) (conjunction analysis), shown with unthresholded maps for (**A**) 5-year-olds and (**B**) 8-year-olds. The underlying data supporting this figure can be found online in files "[Dots2Digits&Digits2Dots] _[5yo/8yo]_LogPval.nii" in "UnthresholdData" folder.
(TIF)

**S7 Fig. Unthresholded maps of between-format decoding across dots and letters.** Brain regions in which activity could accurately classify between adaptation versus no-adaptation blocks of quantity in one format (dots or letters) based on training with the other format (letters or dots) (conjunction analysis), shown with unthresholded maps (for 5-year-olds). The underlying data supporting this figure can be found online in file "Dots2Letters&Letters2Dots _5yo_LogPval.nii" in "UnthresholdData" folder.
(TIF)

**S8 Fig. Results of between-format decoding across digits and letters.** Brain regions in which activity could accurately classify between adaptation versus no-adaptation blocks of quantity

in one format (letters or digits) based on training with the other format (digits or letters) (conjunction analysis). FG, fusiform gyrus; PreCG, precentral gyrus. The underlying data supporting this figure can be found online in files "RawDecAcc_LOOCV_5yo_[Letters2Digits/Digits2Letters]_[Subjects' ID].nii" in "RawDecAcc" folder and in files "Digits2Letters&Letters2Digits_5yo_LogPval.nii" in "UnthresholdData" folder.
(TIF)

**S9 Fig. Differences and similarities in within-format decoding across groups excluding covariates of no-interest.** (**A-B**) Brain regions in which within-group decoding accuracy was larger in 8-year-olds than in 5-year-olds for (**A**) dots and (**B**) digits (based on training with the same format), after regressing out IQ and head motion parameters. (**C-D**) Brain regions in which activity could accurately classify between adaptation versus no-adaptation blocks of (**C**) dots and (**D**) digits in one group based on training with the same format in the other group (conjunction analysis). The underlying data supporting the panels (**A-B**) in this figure can be found in files "RawDecAcc_LOOCV_[5yo/8yo]_[Dots/Digits]_[Subjects' ID].nii" in "RawDecAcc" folder and in files "Regress_[Dots/Digits]_[8yo-5yo/5yo-8yo]_LogPval.nii" in "UnthresholdData" folder. The underlying data supporting the panels (**C-D**) can be found online in files "RawDecAcc_[5to8yo/8to5yo]_[Dots/Digits]_[Subjects' ID].nii" in "RawDecAcc" folder and in files "Regress_[Dots/Digits]_[5to8/8to5]_LogPval.nii" in "UnthresholdData" folder.
(TIF)

**S10 Fig. Between-format decoding using 10-fold cross-validation excluding covariates of no-interest.** (**A**) Brain regions in which activity could accurately classify between adaptation versus no-adaptation blocks of quantity presented in one format (dots or digits) based on training with the other format (digits or dots) (conjunction analysis), after regressing out IQ and head motion parameters. (**B**) Brain regions in which between-format decoding accuracy was larger in 5-year-olds than in 8-year-olds. The underlying data supporting this figure can be found online in files "RawDecAcc_LOOCV_[5yo/8yo]_[Dots2Digits/Digits2Dots]_[Subjects' ID].nii" in "RawDecAcc" folder and in files "Regress_Dots2Digits&Digits2Dots_[5yo/8yo/5yo-8yo/8yo-5yo]_LogPval.nii" in "UnthresholdData" folder.
(TIF)

**S11 Fig. Differences in within-format decoding across groups using 10-fold cross-validation.** (**A-B**) Brain regions in which within-group decoding accuracy was larger in 8-year-olds than in 5-year-olds for (**A**) dots and (**B**) digits (based on training with the same format), using 10-fold cross-validation. Note that the between-groups decoding cannot be performed with this cross-validation method. The underlying data supporting this figure can be found online in files "RawDecAcc_10fold_[5yo/8yo]_[Dots/Digits]_[Subjects' ID].nii" in "RawDecAcc" folder and in files "10fold_[Dots/Digits]_[5yo-8yo/8yo-5yo]_LogPval.nii" in "UnthresholdData" folder.
(TIF)

**S12 Fig. Between-format decoding using 10-fold cross-validation.** (**A**) Brain regions in which activity could accurately classify between adaptation versus no-adaptation blocks of quantity presented in one format (dots or digits) based on training with the other format (digits or dots) (conjunction analysis), using 10-fold cross-validation. (**B**) Brain regions in which between-format decoding accuracy was larger in 5-year-olds than in 8-year-olds. The underlying data supporting this figure can be found online in files "RawDecAcc_10fold_[5yo/8yo]_[Dots2Digits/Digits2Dots]_[Subjects' ID].nii" in "RawDecAcc" folder and in files "10fold_Dots2Digits&Digits2Dots_[5yo/8yo/5yo-8yo/8yo-5yo]_LogPval.nii" in "UnthresholdData" folder.
(TIF)

**S13 Fig. Analyses with the voxel-level threshold of $p < 0.001$.** (**A**) Differences in within-format decoding across groups for (**A**) dots and (**B**) digits adaptation task. (**C-D**) Brain regions in which activity could accurately classify between adaptation versus no-adaptation blocks of (**C**) dots and (**D**) digits in one group based on training with the same format in the other group (conjunction analysis). The underlying data supporting the panels (**A-B**) in this figure can be found in files "RawDecAcc_LOOCV_[5yo/8yo]_[Dots/Digits]_[Subjects' ID].nii" in "RawDecAcc" folder and in files "[Dots/Digits]_[5yo-8yo/8yo-5yo]_LogPval.nii" in "UnthresholdData" folder. The underlying data supporting the panels (**C-D**) can be found in files "RawDecAcc_[5to8yo/8to5yo]_[Dots/Digits]_[Subjects' ID].nii" in "RawDecAcc" folder and in files "[Dots/Digits]_[5to8/8to5]_LogPval.nii" in "UnthresholdData" folder. The underlying data supporting the panels (**E-F**) can be found online in files "RawDecAcc_LOOCV_[5yo/8yo]_[Dots2Digits/Digits2Dots]_[Subjects' ID].nii" in "RawDecAcc" folder and in files "Dots2Digits&Digits2Dots_[5yo/8yo/5yo-8yo/8yo-5yo]_LogPval.nii" in "UnthresholdData" folder.
(TIF)

**S1 Table. Location of brain regions identified in the within-format decoding analysis for dots, differed between 8- and 5-year-olds.** ACC, anterior cingulate cortex; IOG, inferior occipital gyrus; IPL, inferior parietal lobule; IPS, intraparietal sulcus; ITG, inferior temporal gyrus; L, left hemisphere; LG, lingual gyrus; MOG, middle occipital gyrus; MTG, middle temporal gyrus; PoCG, postcentral gyrus; R, right hemisphere; STG, superior temporal gyrus.
(PDF)

**S2 Table. Location of brain regions identified in the within-format decoding analysis for digits, differed between 8- and 5-year-olds.** IFGtri, triangular part of inferior frontal gyrus; MFG, middle frontal gyrus; MFGorb, orbital part of middle frontal gyrus; PreCG, precentral gyrus; SFG, superior frontal gyrus; STG, superior temporal gyrus.
(PDF)

**S3 Table. Location of brain regions identified in the within-format decoding analysis for dots, the similarity between 5- and 8-year-olds (conjunction analysis).** IFG, inferior temporal gyrus; MCC, middle cingulate cortex. MFG, middle frontal gyrus; MFGorb, orbital part of middle frontal gyrus; MTP, middle temporal pole; PreCG, precentral gyrus; SOG, superior occipital gyrus; SPL, superior parietal lobule.
(PDF)

**S4 Table. Location of brain regions identified in the within-format decoding analysis for digits, the similarity between 5- and 8-year-olds (conjunction analysis).** IOG, inferior occipital gyrus; LG, lingual gyrus; MOG, middle occipital gyrus; SOG, superior occipital gyrus.
(PDF)

**S5 Table. Location of brain regions identified in the between-format decoding analysis across dots and digits, by age group (conjunction analysis).** IPL, inferior parietal lobule; PreCG, precentral gyrus.
(PDF)

**S6 Table. Location of brain regions in which accuracy for the between-format decoding analysis across dots and digits, differed between 5- and 8-year-olds.** IFGorb, orbital part of inferior frontal gyrus; IOG, inferior occipital gyrus; IPL, inferior parietal lobule; LG, lingual gyrus; MFG, middle frontal gyrus.
(PDF)

**S7 Table. Location of brain regions identified in the between-format decoding analysis across dots and letters for 5-year-olds (conjunction analysis).**
(PDF)

**S8 Table. Location of brain regions in which accuracy for the between-format decoding analysis across dots and digits was higher than for the between-format decoding analysis across dots and letters for 5-year-olds.** FG, fusiform gyrus; IPL, inferior parietal lobule.
(PDF)

**S9 Table. Location of brain regions identified in the between-format decoding analysis across digits and letters for 5-year-olds (conjunction analysis).** FG, fusiform gyrus; IFGtri, triangular part of inferior frontal gyrus; IOG, inferior occipital gyrus; MOG, middle occipital gyrus; OC, olfactory cortex; PreCG, precentral gyrus.
(PDF)

## Acknowledgments

We thank Franck Lamberton, Danielle Ibarrola, Thomas Bastelica, Jessica Léone, and Justine Epinat for their help with data collection. We are also grateful to the children and parents who participated.

## Author Contributions

**Conceptualization:** Tomoya Nakai, Cléa Girard, Jérôme Prado.

**Data curation:** Cléa Girard, Léa Longo, Hanna Chesnokova.

**Formal analysis:** Tomoya Nakai.

**Investigation:** Tomoya Nakai, Jérôme Prado.

**Methodology:** Tomoya Nakai.

**Project administration:** Jérôme Prado.

**Resources:** Cléa Girard, Léa Longo, Hanna Chesnokova.

**Software:** Tomoya Nakai.

**Supervision:** Jérôme Prado.

**Validation:** Jérôme Prado.

**Visualization:** Tomoya Nakai.

**Writing – original draft:** Tomoya Nakai.

**Writing – review & editing:** Tomoya Nakai, Cléa Girard, Jérôme Prado.

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
