## [Editor Report · Decision Letter 0]

27 May 2022

Dear Dr Nakai, 

Thank you for submitting your manuscript entitled "Cortical representations of Arabic numerals and non-symbolic quantities expand and segregate through early elementary education" for consideration as a Research Article by PLOS Biology.

Your manuscript has now been evaluated by the PLOS Biology editorial staff, as well as by an academic editor with relevant expertise, and I am writing to let you know that we would like to send your submission out for external peer review.

However, before we can send your manuscript to reviewers, we need you to complete your submission by providing the metadata that is required for full assessment. To this end, please login to Editorial Manager where you will find the paper in the 'Submissions Needing Revisions' folder on your homepage. Please click 'Revise Submission' from the Action Links and complete all additional questions in the submission questionnaire. While you are doing this, please also take some time to look over your abstract to ensure that the interesting key findings from your study are clearly evident. This, and the title, are the first aspects of your paper that a reviewer sees. So it is important to ensure they have a good sense of the findings at the outset.

Once your full submission is complete, your paper will undergo a series of checks in preparation for peer review. After your manuscript has passed the checks it will be sent out for review. To provide the metadata for your submission, please Login to Editorial Manager (https://www.editorialmanager.com/pbiology) within two working days, i.e. by May 29 2022 11:59PM.

Kind regards,

Kris

Kris Dickson, Ph.D. (she/her)

Neurosciences Senior Editor/Section Manager

PLOS Biology

kdickson@plos.org

---

## [Decision Letter · Decision Letter 1]

14 Jul 2022

Dear Dr Nakai,

Thank you for your patience while your manuscript "Cortical representations of Arabic numerals and non-symbolic quantities expand and segregate through early elementary education" was peer-reviewed by PLOS Biology. Your work was assessed and discussed by the PLOS Biology editorial team, an Academic Editor with relevant expertise, and by two independent reviewers. Based on the reviews, which you will find at the end of this email, I regret that we will not be pursuing your manuscript for publication at PLOS Biology.

As you will see, there was some discrepancy between the reviewers’ comments but reviewer #1’s comments raised some important concerns that mean that we are not able to consider the manuscript further. R1, Point 8 notes that very different dot and digit numbers are used (6-60 versus 1-8) and this may make interpreting similarities and differences in decoding patterns difficult. R1, Point 9 notes that head motion varies between younger and older groups but is not controlled for; group differences in head motion across can make interpretation of fMRI group differences difficult. R1, Point 4 notes that there is no behavioural evidence regarding the dot or digit discrimination abilities that can inform the neural data. R1, point 3 emphasizes the importance of setting thresholds for significance especially when activity is sometimes reported in quite widespread anatomical areas.

I am sorry that we cannot be more positive on this occasion and hope the reviewer reports will help you in preparing your manuscript for submission elsewhere.

While we cannot consider your manuscript further for publication in PLOS Biology, we would suggest transferring your manuscript, with reviews, to PLOS ONE instead (http://journals.plos.org/plosone/). 

PLOS ONE is a peer-reviewed journal that accepts original research that contributes to the base of academic knowledge. The review process at PLOS ONE focuses on scientific validity, strong methodology and high ethical standards, and the journal's inclusive scope and broad reach means that research published in PLOS ONE will be read, cited and used by researchers across many disciplines. In this case, the PLOS ONE Academic Editors will also take the feedback received from the reviewers at PLOS Biology into account when reaching a decision, which should increase the efficiency of the review process. Please note that the journals are editorially independent and we therefore cannot guarantee the outcome if you choose to pursue a transfer. 

If you would like to submit your work to PLOS ONE, please click the following link:

<DeepLinkData><DeepLinkTypeID>27</DeepLinkTypeID><peopleID>917556</peopleID><userSecurityID>ffd424f9-da36-4cbb-b54e-95be98fdbfc8</userSecurityID><documentID>49839</documentID><revision>1</revision><manuscriptNumber>PBIOLOGY-D-22-01143</manuscriptNumber><docSecurityID>ef8435a1-8699-43f3-82f8-4c9c1df1c302</docSecurityID></DeepLinkData>

If you do NOT wish to submit your work to PLOS ONE, please click this link to decline: 

<DeepLinkData><DeepLinkTypeID>28</DeepLinkTypeID><peopleID>917556</peopleID><userSecurityID>ffd424f9-da36-4cbb-b54e-95be98fdbfc8</userSecurityID><documentID>49839</documentID><revision>1</revision><manuscriptNumber>PBIOLOGY-D-22-01143</manuscriptNumber><docSecurityID>ef8435a1-8699-43f3-82f8-4c9c1df1c302</docSecurityID></DeepLinkData>

Should you choose to transfer your submission to PLOS ONE, you will receive a confirmation email within 24-48 hours of accepting the transfer. Your submission details and manuscript files will be transferred automatically; however, because all PLOS journals vary in submission requirements, once in the PLOS ONE Editorial Manager site, you will be asked to provide additional information before you can finalize your new submission to PLOS ONE. If you have any questions, please feel free to contact the journal at plosone@plos.org.

I hope you understand the reasons for this decision and that the option of publishing your work in PLOS ONE might be useful. Thank you for your support of PLOS and of open-access publishing.

Sincerely,

Roli Roberts

Roland G Roberts PhD

Senior Editor

PLOS Biology

on behalf of

Kris Dickson, Ph.D. (she/her)

Neurosciences Senior Editor/Section Manager

PLOS Biology

kdickson@plos.org

REVIEWERS' COMMENTS:

Reviewer #1:

SUMMARY

Nakai and colleagues report the results of an fMRI study in which 5- and 8-year old children passively viewed dots or digits. Their two main findings are smaller cortical responses in 5yos vs. 8yos and a format-independent response to dots and digits in 5yos (parietal cortex) but not in 8yos. The authors conclude that their results can be explained by "the symbolic estrangement hypothesis, which argues that the relation between symbolic and non-symbolic quantity weakens with exposure to formal mathematics in children". 

EVALUATION

Strengths:

-multivariate analysis approach

Weaknesses:

-cross-sectional study design

-voxel-wise decoding accuracy is not reported

-reliability of multiple comparison correction is unclear

-lack of behavioral performance data for numerosity detection

-neuroanatomical localization is intransparent

-conclusions about group differences are at least in part not statistically validated 

-differences in numerosity detection could also be differences in other cognitive domains not tested

-numerosities are not matched between formats

-differences in head motion are not modeled

MAJOR CONCERNS

1. Introduction, l.105-107: Decoding fMRI adaptation patterns using training data of 5yos and held-out independent test data of 8yos is statistically valid. But given the cross-sectional design of the present study the results reported here cannot answer the question whether neural representations of numerosity are stable (or shrink or expand) through the first four years of elementary education. My suggestion is to address this limitation to the discussion section and explicitly state that longitudinal work is needed to answer this question.

2. Results, Fig.2: The decoder reveals that almost the entire neocortex of 8yos contributes to classifying adaptation vs. no-adaptation dots. I do not find this neurobiologically plausible since I am not aware of literature providing evidence for visual numerosity coding in body motor and somatosensory cortex or auditory cortices. Therefore it would be highly desirable to report a statistical map showing the whole-brain voxel-wise decoding accuracy of the searchlight. Please note that this concern also applies to the other figures. 

3. Methods, l.567-572: In cluster-based multiple comparison correction an uncorrected cluster-forming voxel-wise threshold of P < 0.005 is considered as too liberal to ensure robust replication (Eklund et al. 2016 PNAS). Please provide a reference proving that this problem does not arise in permutation-based multiple comparison correction or apply conservative voxel-based thresholding (family-wise error or false discovery rate). 

4. Results, general: I consider it as a problem that it is not reported how well the participants can identify and discriminate dots and digits at the behavioral level. I find it unrealistic to expect that age is perfectly correlated with numerosity detection performance given that there are pronounced individual differences in math ability. My recommendation is to address this limitation in the discussion section. I would state that the results reported here do not allow to draw conclusions about numerosity detection ability. 

5. Results, general: In my view, there is a mismatch between the surface renderings of the classification accuracy maps and the neuroanatomical labels assigned by the authors. For example, in Fig. 3B, L.DLPFC seems to be the premotor cortex instead. Please provide MNI coordinates and use an atlas to assign neuroanatomical labels.

6. Results, Fig.4: I think that the authors cannot infer a group difference from a significant finding in one group (5yo) and a null finding in the other group (8yo). To make this claim they would have to report a statistical index showing that the decoding accuracy in one group is significantly higher. Is this the case? 

7. Results, l.254-267: While attention did not differ within each age group, I think that the authors cannot rule out that the different fMRI patterns might not be explained by between-group differences in numerosity detection, but by between-group differences in attention. Moreover, the fronto-parietal cortices house several other systems (incl. working memory and metacognition (theory of mind)) so that differences in these domains are plausible alternative explanations that are not controlled here. I consider it as very important to acknowledge this substantial limitation in the discussion. Also, although IQ differences between the age groups did not reach significance it would nevertheless be desirable to include Iq scores as covariates of no interest in the models to control for an effect of general cognitive ability.

8. Methods, Adaptation task: I find it problematic that the number of dots (6-60) did not match the numerosities corresponding to the digits (1-8). I fear that these differences wrt the numerosities could explain the lack of format independence in the older children. Please discuss this in extenso in the discussion.

9. Methods, fMRI data preprocessing: Surprisingly, more 8yos than 5yos were excluded due to head motion and head motion was still higher in the remaining 8yos. Please include head motion indices as covariates of no interest in all models. 

MINOR POINTS

1. General remark: The manuscript would benefit from proofreading by a native English speaker. There are a number of semantic and grammatical mistakes as well as stylistic flaws in the text.

2. Results, general: Are the results robust to the cross validation procedure chosen here? Please provide the results obtained using 10-fold cross validation (Knops et al. 2009).

3. Discussion, l.322-325: I do not understand why larger numerosity responses in 8yo vs 5yo are consistent with the idea that the acquisition of symbolic number skills refine the cognitive mechanisms supporting non-symbolic quantity processing. Please clarify.

4. Discussion, l.378-379: I do not understand why the fact that the authors observed between-format decoding in both hemispheres can be explained by the fact that orthographic features are shared by both digits and letters. Please explain.

5. Methods, Participants: Were the 49 children that dropped out during fMRI comparable to the 89 children that did not drop out in terms of their cognitive ability and socioeconomic background?

6. Methods, Behavioral testing: It should be mentioned that the children enrolled in the present study had above average intelligence. Could this be explained by a selection bias? 

7. Methods, l.510-520: I do not understand how anatomical differences between children older than 5- to 8-years and adults relate to the current sample. Please clarify.

8. References: Formatting is inconsistent. Please double-check.

REFERENCES

Eklund et al. (2016) PNAS https://doi.org/10.1073/pnas.1602413113

Knops et al. (2009) Science DOI: 10.1126/science.1171599

Reviewer #2:

The paper is well written, and the study is well-motivated and situated in the existing literature.

Overall, the analyses are appropriate to the theoretical questions, nicely structured and clearly reported.

I think this a well-done, interesting study that makes a valuable contribution to the literature.

I only have one question which is, regarding the adaptation paradigm, isn't it more traditional to model the deviant trials as events, rather than model the entire block? Could the authors expand on this methodological decision and briefly discuss the potential impact on results? This question is not critical to the acceptance of the paper in my opinion.

---

## [Editor Report · Decision Letter 2]

28 Jul 2022

Dear Dr Nakai,

Thank you for your patience while your appeal on the manuscript "Cortical representations of Arabic numerals and non-symbolic quantities expand and segregate through early elementary education" was evaluated by the PLOS Biology editors and our Academic Editor. 

In light of your comments and our discussion, we are rescinding the original rejection decision and would like to invite you to revise the work to thoroughly address the reviewers' reports.

We would like to thank you for clarifying the nature of the permutation-based test that was used (point 1). It would be advisable to emphasize this point clearly so that there is no possibility of misunderstanding when the study comes back in for re-review. We also ask that you include some of the behavioural evidence that is mentioned in point 2. While we understand that the neuroimaging analysis may have been performed on tasks that can be performed similarly by both age groups it would be useful to put those results in context by showing additional behavioural data from the participants even if they were not collected at the time of the neuroimaging experiment. As for point 4, despite the awkward phrasing, our interpretation is that this is reflects concern that when a comparison is made between groups of different participants then care needs to be taken that there are no differences in head movement (for example, Power, Schlagger, Petersen, NeuroImage, 2015). We therefore ask that you consider some of the strategies that have been suggested for controlling for the impact of head motion differences.

Given the extent of revision needed, we cannot make a decision about publication until we have seen the revised manuscript. Your revised manuscript is likely to be sent for further evaluation by all or a subset of the reviewers.

**IMPORTANT - SUBMITTING YOUR REVISION**

*Re-submission Checklist*

*Published Peer Review*

*PLOS Data Policy*

*Blot and Gel Data Policy*

Sincerely,

Kris

Kris Dickson, Ph.D. (she/her)

Neurosciences Senior Editor/Section Manager

PLOS Biology

kdickson@plos.org

REVIEWS:

---

## [Decision Letter · Decision Letter 3]

28 Oct 2022

Dear Dr Nakai,

Thank you for your patience while we considered your revised manuscript "Cortical representations of Arabic numerals and approximate non-symbolic quantities expand and segregate through early elementary education: A cross-sectional comparison between children aged 5 and 8" for consideration as a Research Article at PLOS Biology. Your revised study has now been evaluated by the PLOS Biology editors, the Academic Editor and the two original reviewers. 

As you will see from the reviewer comments, there is still a split in views with Reviewer 1 not supportive and Reviewer 2 supportive. After discussion with our Academic Editor, we feel it would be appropriate to move forward with this work at PLOS Biology provided you make some additional changes to ensure that our readership can critically evaluate this work and come to their own conclusions. In this regard, we ask that you include the following in the final revised version of this work:

1) Please ensure that all summary data is deposited in a publicly available, unrestricted, repository (e.g. Zenodo) as per our journal policy. The current datasets you provided are restricted.

2) Please also provide whole-brain voxel-wise decoding accuracy maps in the Zenodo repository, in addition to the provided p-value maps, so that readers can determine for themselves which voxels contribute most strongly to maximum decoding accuracy.

3) Reviewer 1 continues to feel that reanalyzing the data with a p <0.001 would make for stronger arguments (their point 2). While appreciating the counter arguments you’ve put forth, in discussing this further with Reviewer 2, there was some agreement that this tighter threshold would have given more confidence in your results. We’d therefore encourage you to consider addressing this concern with additional analysis. Minimally, we ask that you provide a more extensive discussion of your reasoning for the choice you’ve made in the main text of your study.

4) Please clearly acknowledge that children with lower IQs were excluded from your analysis and provide a more detailed discussion of why this was done. We appreciate that such exclusions are often necessary given issues with attention, task abilities, etc. Nevertheless, we feel that this point, and any potential implications from this, should be clearly made in the text of your work. 

Please also make sure to address the following data and other policy-related requests.

**Data Policy – 

Note that we do not require all raw data. Rather, we ask that all individual quantitative observations that underlie the data summarized in the figures and results of your paper be made available. 

The scripts and individual MRI data available from Zenodo (http://doi.org/10.5281/zenodo.6472638) are currently restricted. This will need to be updated to allow free and open access.

**Figures/Figure legends - 

Please ensure that you provide the individual numerical values that underlie the summary data displayed in the following figure panels as they are essential for readers to assess your analysis and to reproduce it:

Fig 2E,F; Fig 3E,F; Fig 4E,F; Fig 5C

Supp Figs: data underlying heatmaps for Supp Figs 1-12. 

Please also ensure that figure legends in your manuscript include information on where the underlying data can be found (e.g. “The underlying data supporting Fig X, panel Y can be found in file Z.”)., and ensure your supplemental data file/s has a legend.

**Blurb – 

Please also provide a blurb which, if the paper is accepted, will be included in our weekly and monthly Electronic Table of Contents (eTOCs), sent out to readers of PLOS Biology. This blurb may also be used to promote your article on social media. The blurb should be about 30-40 words long and is subject to editorial changes. It should, without exaggeration, entice people to read your manuscript, should not be redundant with the title and should not contain acronyms or abbreviations. For examples, view our author guidelines: https://journals.plos.org/plosbiology/s/revising-your-manuscript#loc-blurb

**IMPORTANT - SUBMITTING YOUR REVISION**

*Published Peer Review History*

*Press*

Sincerely,

Kris

Kris Dickson, Ph.D., (she/her)

Neurosciences Senior Editor/Section Manager

PLOS Biology

kdickson@plos.org

REVIEWS:

Reviewer's Responses to Questions

PLOS authors have the option to publish the peer review history of their article (what does this mean?). If published, this will include your full peer review and any attached files.

Reviewer #1: No

Reviewer #2: No

Reviewer #1: I would like to thank the authors for trying to address the concerns that I have raised in my previous review. Nevertheless, I am convinced that a number of substantial weaknesses remain which I will take up in detail below. 

(1) Lack of neurobiological validity: A searchlight decoder is agnostic in terms of neurobiological validity. It just goes through each voxel and tries to solve a given decoding problem. This is why the current finding that visual numerosity is represented in auditory and body motor cortex requires neurobiological validation. But even the most powerful ultra-high-field fMRI study by Harvey et al. 2017 mentioned by the authors did not detect any visual numerosity representations in auditory and body motor cortices. Let alone any direct neuronal recording work (Nieder 2016). I find this really problematic because fMRI is prone to false-positive findings. 

(2) Lack of statistical explicitness and transparency: Given the moderate decoding accuracy reported here (in the range of 45-60%) I consider it as very important to make whole-brain voxel-wise decoding accuracy maps available to the reader (as I have asked in my previous review). Right now, it is impossible to determine which voxels contributed most strongly to the maximum decoding accuracy. The p-value maps buried in the Supplement do not contain this crucial information. In addition, having a closer look at these p-value maps (Fig.S1S2), I find it very concerning to see that p-values are highest in the canonical frontoparietal number system and lowest in frontal and occipital poles!? 

(3) Liberal multiple comparison correction: It is unclear to me why the authors stick to their liberal cluster-forming threshold of p < 0.005 instead of reanalyzing the data using the the well-established threshold of p < 0.001 (see e.g. Cavdaroglu & Knops 2019). To ensure replicability by other labs, I see it as essential to provide direct empirical evidence based on the present dataset that the permutation test chosen here robustly removes false-positive activation across cluster-forming thresholds. 

(4) Statistical power unknown: Sample size selection is not guided by an a priori power analysis so that actual statistical power is not known.

(5) Behavioral relevance: The present work does not speak to the important question of whether the reported neural data explain behavior (Krakauer et al. 2017). It remains open whether neural representation metrics relate to individual mathematical test performance scores. 

(6) Mismatch between stimuli: Dots did not match digits in the current task (6-60 vs. 1-8). Comparing these conditions would thus be comparing apples and oranges. Accordingly, the authors justifiably did not directly compare dots and digits. But this also means that (for experimental reasons) their data do not support their interpretations about the association between symbolic and non-symbolic quantity. 

(7) Selection bias: I find it severely misleading to label an average IQ of 109 or even 112 as "slightly above the norm". Instead, families with above-average intelligence were significantly overrepresented here. This problem is aggravated in my view by the fact that the 49 excluded children (about 40% of the entire sample) were less intelligent (p = 0.06) than the 89 children who were retained for MRI scanning. As a consequence, I see no empirical basis for generalizing the reported findings to general population. 

My conclusion is that the remaining weaknesses limit the potential impact of the work on the field considerably and compromise the utility of the methods and data to the community.

References:

Cavdaroglu & Knops 2019 Cerebral Cortex https://doi.org/10.1093/cercor/bhy163

Harvey et al. 2017 Nature Human Behavior https://doi.org/10.1038/s41562-016-0036

Krakauer et al. 2017 Neuron https://doi.org/10.1016/j.neuron.2016.12.041

Nieder 2016 Nature Reviews Neuroscience https://doi.org/10.1038/nrn.2016.40

Reviewer #2: I had no suggestions in my prior review. I think the manuscript is still excellent and acceptable for publication in its current form

---

## [Editor Report · Decision Letter 4]

30 Nov 2022

Dear Dr Nakai,

Thank you for the submission of your revised Research Article "Cortical representations of Arabic numerals and approximate non-symbolic quantities expand and segregate through early elementary education: A cross-sectional comparison between children aged 5 and 8" for publication in PLOS Biology. We appreciate the additional revisions that have been made to this work, including reanalyzed the data at a more stringent p value (<0.001) as Reviewer 1 had requested and providing all of your raw data so that others can critically evaluate your work. Given these additions, on behalf of my colleagues and the Academic Editor, Matthew Rushworth, I am pleased to say that we can in principle accept your manuscript for publication. 

***We do ask that you consider a title change to this work to make the findings more broadly accessible. Please consider one of the following:

Cortical representations of numbers and non-symbolic quantities expand and segregate in children from 5 to 8 years of age

A cross-sectional analysis demonstrates both expansion and segregation of cortical representations for numbers and non-symbolic quantities in children from 5 to 8 years of age

***We also ask that you address any remaining formatting and reporting issues. These will be detailed in an email you should receive within 2-3 business days from our colleagues in the journal operations team; no action is required from you until then. Please note that we will not be able to formally accept your manuscript and schedule it for publication until you have completed any requested changes.

PRESS

Sincerely, 

Kris

Kris Dickson, Ph.D., (she/her)

Neurosciences Senior Editor/Section Manager

PLOS Biology

kdickson@plos.org